# Primary School Pupils: Unequal GMC Developmental Pathways in a Single School Year

**DOI:** 10.3390/children9070964

**Published:** 2022-06-28

**Authors:** Mark de Niet, Veerle M. A. Wetzels, Johan Pion, Irene R. Faber, Sebastiaan W. J. Platvoet, Marije T. Elferink-Gemser

**Affiliations:** 1Department of Sports and Exercise, HAN University of Applied Sciences, 6525 AJ Nijmegen, The Netherlands; johan.pion@han.nl (J.P.); sebastiaan.platvoet@han.nl (S.W.J.P.); 2Department of Human Movement Sciences, University Medical Center Groningen, University of Groningen, 9713 AV Groningen, The Netherlands; v.m.a.wetzels@student.rug.nl (V.M.A.W.); m.t.elferink-gemser@umcg.nl (M.T.E.-G.); 3Department of Movement and Sports Sciences, Faculty of Medicine and Health Sciences, University of Ghent, 9000 Ghent, Belgium; 4Institute of Sport Science, University of Oldenburg, 26129 Oldenburg, Germany; irene.faber@uni-oldenburg.de; 5Research Centre Human Movement and Education, Windesheim University of Applied Sciences, 8017 CA Zwolle, The Netherlands

**Keywords:** motor development, KTK3+ test, individual development, physical education

## Abstract

Gross motor coordination (GMC) is essential for the development of specific motor skills and long-term participation in physical activities and sports. Group analysis reveals that, on average, children develop these skills gradually; however, how individuals develop GMC is less clear. The main aim of this study is to increase the understanding of developmental patterns within one school year, and whether children’s grade, gender, or baseline GMC proficiency are associated with these developmental patterns. In total, 2594 Dutch children aged 6–12 years performed the modified Körper Koordinations Test für Kinder (KTK3+) twice in one school year (autumn and spring). The KTK3+ includes four subtests: walking backwards, moving sideways, jumping sideways, and eye-hand coordination (EHC) test. On average, children developed significantly on all subtests (*p* < 0.001). At baseline, children in higher grades scored significantly higher than children in lower grades, and children in grades 5 and 6 (age 9 and 10 years) showed most development (raw scores on average, *p* < 0.001). Boys outperformed girls on EHC across all grades, whereas girls outperformed boys on walking backwards. Nevertheless, both boys and girls developed similarly. Children with lower scores at baseline developed more quickly across all grades. Noteworthy is that 12.1–24% (depending on the test item) of the children scored lower in the spring than in the autumn tests. On average, children develop their GMC; however, we report large differences in their individual trajectories and note that a substantial number did not show a positive GMC development. Further research should examine GMC development with more possible influencing factors as well as over a longer time span to better understand differences in children’s GMC development. This may result in more individualized programs in PE lessons, enabling children to optimally develop their GMC, and better use of GMC assessment tools to monitor children’s development.

## 1. Introduction

Gross motor coordination (GMC) is essential to the development of specific motor and sport competences [1,2]. It is associated with long-term involvement in physical activity [3] and future sport competition levels [4]. Especially from the ages 6 to 12 years, it is assumed that children develop their GMC rapidly, mainly due to an acceleration of cortex maturation and to the cortex becoming more organized [5,6]. As most primary school children aged 6–12 years have physical education (PE) classes twice a week in the Netherlands, it is unsurprising that PE teachers play an important role in the development of children’s GMC [7,8]. A good understanding of a child’s individual GMC development, both short-term and long-term, may help PE teachers to better meet individual children’s developmental demands and evaluate their PE programs. In general, children’s GMC does develop over time, but the rate of development varies between children [9]. This increases the need for PE teachers to monitor GMC proficiency development. However, little research has been conducted on individual GMC development in a relatively short-time period (e.g., one school year). As most PE teachers evaluate and adjust their programs yearly, a better understanding of individual GMC developmental patterns may increase the quality of PE.

In recent years, PE teachers have started to use popular and relatively easily applicable PE tests to assess GMC, such as the modified Körper Koordinations Test für Kinder (KTK3+) [10,11]. PE teachers most often use reference scores to interpret the children’s GMC. Research has shown that children aged 6 to 12 years positively develop their GMC over time, i.e., on average, a 7-year-old performs better than a 6-year-old [2,12,13,14]. However, the rate of development of GMC decreases over time [12]. Professionals working with young children might therefore assume that children improve their GMC within one school year, as this is supported by increases in reference scores. Recent studies however have shown that there are large differences in children’s individual GMC development, and that some may even have limited or no development [12,15]. A limited understanding of these individual developmental differences decreases the value of using group references for PE teachers. Therefore, to better meet a child’s developmental demands, insights into each individual GMC proficiency would be more meaningful for a PE teacher than the average GMC proficiency of a particular grade or age.

Several factors may be related to the developmental trajectories, including gender, age [[8],[12], and current GMC proficiency [12,16]. Studies have shown gender differences in GMC proficiency, where boys outperform girls on object control tasks, while girls outperform boys on balancing tasks [10,17,18]. These differences may be explained by gender differences in play: boys play with a ball more, and girls do activities like rope jumping and other balancing tasks related to gymnastics. However, contrasts have been shown regarding differences between boys and girls in their rate of GMC development. One study showed that boys have a higher rate of GMC development over time than girls [12], whereas another study showed that girls showed more FMS development than boys [15]. In contrast to these studies, Rodrigues et al. (2016) [19] and Platvoet et al. (2016, 2020) [8,14] found no gender differences in the rate of GMC development.

Different results were also found regarding the influence of baseline GMC proficiency on GMC development. Research showed no clear association between GMC proficiency at baseline and the rate of GMC development in two years [12]. In line with these results, Platvoet and colleagues (2020) [14] reported that children with higher sport learning capacity developed greater GMC over 24 weeks than those classified with lower sport learning capacity. These results contrast those reported by Dos Santos and colleagues (2018) [15] who showed that higher GMC proficiency of six-year-old children at baseline was associated with a lower GMC development rate over four years, which may suggest a ceiling effect in GMC proficiency. These discrepancies in the literature reveal that the association between GMC proficiency and the development of GMC proficiency is unclear.

As stated, data on individual children’s GMC development in one primary school year (in the Netherlands) are still relatively scarce. Therefore, the aim of this study is to increase our understanding of these developmental patterns. To achieve this, we will first determine the level of a child’s GMC proficiency and individual development in one school year, followed by determining whether this development is influenced by children’s grade, gender, and baseline proficiency. We hypothesize that most children, regardless of grade, gender, and baseline proficiency, will develop their GMC within one school year, but that the rate of development between children varies. Furthermore, we hypothesize that children in higher grades (older) will, on average, develop their GMC more slowly than children in lower grades (younger). Due to the contrasting results in the literature on the influence of gender and current developmental proficiency levels, we cannot hypothesize whether these factors either positively or negatively influence children’s GMC development.

## 2. Materials and Methods

### 2.1. Participants

A total of 2594 Dutch children (1247 boys and 1347 girls) participated in this study from 21 primary schools in the Netherlands. Ten schools were located in two average-sized cities (Arnhem and Nijmegen), and eleven schools in a large city (The Hague). The participants ranged from the third grade (age 6.5 ± 0.5 years) to the eighth grade (age 11.6 ± 0.5 years) according to the Dutch school system. All children had PE from a qualified PE teacher twice a week for 45 min. The content of PE was based on the Dutch Handbook for PE [20]. Children who were unable to complete the assessment (due to injury or other physical limitations) were excluded from this study. Parents/caretakers of each child were informed at the start of the study. If a child did not want to participate, or was not allowed by their parents/caretaker to participate, data were not collected on the specific child. The study was approved by the ethical advisory committee at the Faculty of Health of the HAN University of Applied Sciences (reference number EACO 17.12/89).

### 2.2. Gross Motor Coordination Assessment

To assess GMC, the participants performed the KTK3+, which consists of three items of the Korper Köordinatons Test für Kinder (KTK): walking backwards (WB, three attempts on three different sized beams, total of nine attempts, score is total number of steps with a maximum of 72), moving sideways (MS, two attempts of 20 s, score is total number of sideway moves), jumping sideways (JS, two attempts of 15 s, score is total number of sideway jumps) [21], as well as an eye hand coordination (EHC, two attempts of 30 s, score is total number of catches) test item [22]. The KTK3+ measures the three components of GMC (locomotion, balance/stability, and object control) and is feasible and regularly used by PE teachers in the Netherlands. The test-retest reliability of the test items is considered good (WB 0.80, MS 0.84, JS 0.95, EHC 0.87 [21,22]) as is the validity [11]. The test was assessed by undergraduate PE students in their last year of education. They were trained to assess the test according to the protocol (described in the study of its development [10]). The test was performed twice (T0 in autumn, approximately 6–8 weeks after the start of the school year and six months later (T1) in spring, approximately 6 weeks before the end of the school year) by the same participants during their regular physical education class.

### 2.3. Data Analysis

IBM SPSS Statistics 27 (IBM Corp., Armonk, New York, NY, USA) was used for all the statistical analyses. First, children’s baseline proficiency levels were determined per test item. The 15% lowest scores on each test item (per grade and gender) were classified as low, with the 15% highest scores as high. All other scores were classified as average. Second, the mean and standard deviation of the test item scores on T0 and T1 were calculated for each test item (both overall and per grade and gender). Third, repeated measures ANOVA was used to elucidate both the main and interaction effects of grade, gender, and baseline proficiency, the three between-subject factors, and the within-subjects variable time per test item. In case of significant main and/or interaction-effects, post hoc analysis was conducted with independent samples *t*-test, with Bonferroni correction. Fourth, to determine whether the children within the three proficiency level categories at baseline scored significantly differently from each other at T0 and on T1, independent samples *t*-tests with Bonferroni correction were performed.

The EHC-test item was conducted differently for children in grades 3 and 4 (allowed to catch the ball with two hands) compared to children in grade 5–8 (they had to catch the ball with one hand) [10]. Due to this difference in the protocol, results of the EHC-test cannot be compared between grades. Therefore, only results of the EHC-test item of children in grades 5–8 were used for analysis.

Based on the smallest detectable difference (SDD) of the test items, it was determined whether children showed positive (T1-T0 ≥ +SDD), no (-SDD < T1-T0 < +SDD), or negative (T1-T0 ≤ −SDD) development. The SDDs for the four test items were based on previous test-retest studies within two weeks (2.7 (WB), 2.5 (MS), 5.7 (JS) [21]; and 2.7 (EHC) [22]). Frequencies of positive, no, and negative development were calculated for each subtest per grade, for gender, and for baseline proficiency level.

Alpha was set at 0.05 for statistical significance for all analyses. For the repeated measures ANOVA, the partial eta squared was calculated. The partial eta squared is interpreted as follows: *η2* = 0.01 indicates a small effect, *η2* = 0.06 indicates a medium effect, and *η2* = 0.14 indicates a large effect [23].

## 3. Results

The repeated measures ANOVAs showed that, on average, children scored significantly higher on all test items at T1 compared to T0 (*p* < 0.001, WB: F(1,2558) = 383.911; partial *η^2^* = 0.130, MS: F(1,2558) = 350.842; partial *η^2^* = 0.121, JS: F(1,2558) = 528.822; partial *η^2^* = 0.171, EHC: F(1,1766) = 623.067; partial *η^2^* = 0.261, see Figure 1, Figure 2, Figure 3 and Figure 4 and descriptive statistics in Appendix A). Of the children, 47.5–63.5% showed positive development, 17.3–40.4% showed no development, and 12.1–24.0% showed negative development for all four tests (ranges over four test items, see Appendix A and Figure 5).

### 3.1. Grade

In the repeated measures ANOVA, the between-subjects effect ‘grade’ was significant for all four test items (all: *p* < 0.001, WB: F(5,2558) = 304.462; partial *η^2^* = 0.373, MS: F(1,2558) = 652.610; partial *η^2^* = 0.561, JS: F(1,2558) = 819.656; partial *η^2^* = 0.616, EHC: F(1,1766) = 309.249; partial *η^2^* = 0.334). On average, children scored significantly better than their one grade lower peers for all four test items (all comparisons: *p* < 0.001), except for WB between grades 6-7 (*p* = 0.094).

For all test items, an interaction-effect between time and grade was found (WB: F(5,2558) = 6.011; *p* < 0.001; partial *η^2^* = 0.012, MS: F(5,2558) = 7.189; *p* < 0.001; partial *η^2^* = 0.014, JS: F(5,2558) = 2.496; *p* = 0.029; partial *η^2^* = 0.005, EHC: F(3,1766) = 11.231; *p* < 0.001; partial *η^2^* = 0.019). For WB, children in grades 5–6 developed, on average, more than children in grades 3 and 8, and children in grade 5 also developed more than children in grade 4 (see Appendix A). For MS, children in grade 3 showed significantly less development than children in other grades (see Appendix A). For JS, children in grade 6 showed significantly more development than children in grades 3, 4, and 8 (4 ± 10, *p* < 0.001, see Appendix A). Finally, for EHC, children in grade 5 (11 ± 11) showed significantly more development than children in the other grades (see Appendix A). All other comparisons between grades for the four test items were not significant.

### 3.2. Gender

We found no difference in development between boys and girls (no interaction effect between time and gender; WB: F(1,2558) = 0.182; *p* = 0.670, MS: F(1,2558) = 1.396; *p* < 0.238, JS: F(1,2558) = 0.058; *p* = 0.810, EHC: F(1,1766) = 1.040; *p* = 0.308).

The main effect of gender was significant for WB, JS, and EHC (for these test items: *p* < 0.001, WB: F(1,2558) = 220.267; partial *η2* = 0.079, JS: F(1,2558) = 10.105; partial *η2* = 0.004, EHC: F(1,1766) = 226.139; partial *η2* = 0.114), while there was no significant difference in proficiency level between boys and girls on MS (F(1,2558) = 2.104; *p* = 0.147). Boys outperformed girls for JS (test score boys: 59 ± 16, girls: 58 ± 16) and EHC (boys: 25 ± 14, girls: 18 ± 13), whereas girls outperformed boys on WB (boys: 35 ± 13, girls: 41 ± 13).

### 3.3. Baseline Proficiency Level

On average at T0, the children in the three proficiency level categories showed significant performance differences from each other for all tests (all comparisons: *p* < 0.001). The interaction effect between time and baseline score was significant for all test items, as showed by the repeated measures ANOVA (for all: *p* < 0.001, WB: F(2,2558) = 176.610; partial *η2* = 0.121, MS: F(2,2558) = 165.594; partial *η2* = 0.115, JS: F(2,2558) = 91.687; partial *η2* = 0.067, EHC: F(2,1766) = 92.791; partial *η2* = 0.095). For all test items, children in the ‘low scores’ category (T1-T0: WB = 11 ± 10, MS = 7 ± 7, JS = 10 ± 11, EHC = 9 ± 10) developed more than children in the ‘average scores’ (T1-T0: WB = 5 ± 10, MS = 3 ± 7, JS = 5 ± 9, EHC = 8 ± 10) and ‘high scores’ category (T1-T0: WB = −2 ± 10, MS = −1 ± 7, JS = 1 ± 10, EHC = 1 ± 10). Children in the ‘average scores’ category showed more development than those in the ‘high’, and categories performed significantly different from each other for all tests at T1 (for all comparisons *p* < 0.001, i.e., ‘low score’ children scored, on average, lower than ‘average’ and ‘high’ children).

## 4. Discussion

The aim of this study was to increase our understanding of children’s individual GMC developmental trajectories in one school year in primary schools, and whether individual development is influenced by grade, gender, and baseline proficiency. The results show that most of the children developed their GMC. However, a substantial group of children did not. Both grade and baseline GMC proficiency were associated with children’s development, as children in grades 5 and 6 showed (on average) more GMC development than other grades. Those with a lower baseline GMC proficiency developed their GMC more (on average). No differences in development were found between boys and girls.

A significant percentage of the children showed no improved development (17.3–40.4%) and 12.1–24.0% even had a negative development curve in one school year. The development of motor skills is thus a non-linear trajectory [24] which may explain the significant number of children with no or negative development in raw scores in a shorter time frame (one school year). However, previous studies have shown that almost all children will develop (i.e., 99.6% in the study of Coppens and colleagues (2019) [12], and Rodriques and colleagues (2016) [19]) GMC over a longer time frame (2–4 years). In line with these studies, our results show an overall increase in GMC proficiency over grades. Based on the study by Coppens and colleagues (2019) [12], it is reasonable to assume, and therefore important for PE teachers to understand, that most of the children who did not improve themselves in one school year will do so in a higher grade.

In line with Dos Santos and colleagues (2018) [15], children’s baseline GMC proficiency influenced their development in our study, regardless of grade and gender. Those with lower proficiency levels developed more than those with a higher proficiency level at baseline, even in the higher grades. Additionally, after one school year, proficiency levels of the children in the ‘low scores’ group were higher than baseline GMC proficiency levels of one grade higher (e.g., at WB, boys in the low proficiency group scored 18 ± 7 at T1 in grade 3 compared to 11 ± 2 at T0 in grade 4). Although this was not based on longitudinal data, it suggests that especially children with low proficiency levels may be at risk of a fallback after the summer holidays. An explanation might be found in the importance of a structured environment (i.e., school) for the development of GMC for those with lower proficiency levels who also often grow up in a lower social-economic environment and have lower sport participation rates [25]. Children with low GMC proficiency may be less (stimulated to be) physically active outside the school environment (e.g., in sports) while those with average and especially high GMC proficiency play more sports and are more physically active outside school and during school holidays. For those children with a lower GMC proficiency, PE may be essential for their GMC development.

A complementary or alternative explanation for the negative association between development and baseline GMC proficiency may be that PE teachers focus more on children with less developed GMC. This may improve these children’s development, while more proficient children may not be challenged enough and therefore develop relatively less [14]. In most PE handbooks, less information or activities are available to challenge the most gifted children. In a class with over 25 children, it is often difficult to differentiate between those who already show the required proficiency for their specific grade. However, the more ‘gifted’ should be given the same attention and opportunities to develop themselves [14,26]. Finally, it could be more difficult for the more ‘gifted’ children to develop their GMC as they start when they already more proficient. Our study shows that children in subsequent grades scored significantly higher than their peers in lower grades. This suggests that children still have room for improvement, even in higher grades, though some children may have reached their ceiling in that specific grade. As we did not follow children’s development over grades (more years) and do not have information about children’s sport participation, social economic status, or quality of PE classes, more research, over a longer time period and with intermittent data collection, is warranted to better understand how proficiency levels relate to children’s GMC development.

Based on Coppens and colleagues (2019) [12] and the “mountain of motor development” model [1], we expected that children in the lower grades (i.e., 3–5) would show the most improvement in GMC and that the rate of development would decrease with age. Our results show that all grades improve on average, but that children in grades 5 and 6 (age: 8–10 years old) showed, relatively, the most development. An explanation might be that children in grades 5 and 6 are both physically and cognitively optimally developed to improve their GMC compared to their younger peers [9]. After grade 6 (in general), the development of sport-specific skills becomes even more important, resulting in less focus on GMC development. The GMC can still improve, but most children improve at a lower rate. Our results show that children in grades 7 and 8, i.e., 10–12-year-olds, in the low proficiency group showed large developments. Currently, a substantial number of eight-year-old children have low GMC proficiency compared with previous decades (e.g., [2,27]). This also suggests that we should critically look at our current PE curricula. For example, in line with the mountain of motor development and PE handbooks, PE classes for children aged 9 or 10 focus more on context-specific motor skills (i.e., sports). However, children whose GMC proficiency is insufficient for these context-specific motor skills (those who hit the proficiency barrier) will face difficulties in participating in these context-specific activities (e.g., a child not being able to throw and catch a ball in a more closed situation will experience more difficulties in a dynamic game like oddball). Therefore, we strongly recommend critically reviewing our PE curricula, with individual GMC proficiency becoming more prominent in activity choice than using activities designed for specific age-groups.

Finally, in line with previous studies [10,17,18], we found gender differences in proficiency, but not in the rate of development. Boys outperformed girls on object control, whereas girls outperformed boys on balancing tasks. These findings are consistent with those of several studies [8,14,19]. One explanation may be that boys and girls practice different activities in leisure time, resulting in more object control for boys and balancing skills for girls [28,29]. However, this may not completely explain the difference in GMC proficiency, as, in our study, the rate of development did not differ between boys and girls. Although previous studies report different results in rate of development, boys had a greater rate of development than girls [12] and girls made more progress than boys over time [15]. Our results suggest that this difference in balancing and object control proficiency already appears before the third grade and persists across the grades. This may be a consequence of genetic predispositions [28]. However, further research is needed to gain more insight into the causes of the difference in different aspects of GMC between boys and girls over time.

The main strengths of this study are its large sample size and the broad range of grades. Comparable studies from Dos Santos et al. (2018) [15] and Coppens et al. (2019) [12] had significantly smaller samples sizes (all between 200 and 500 participants) and a smaller range of ages. With 2594 children, this study had a representative sample for the Dutch primary school population from grades 3–8. However, the first limitation is that the KTK3+ test does not measure all GMC aspects extensively (e.g., walking and kicking). When using the KTK+3 test, all three categories of GMC performance can be assessed in a limited time [10]. Secondly, we did not measure what children did during PE classes and outside school (i.e., playing outside, playing sports) between the first and second measurement. All participating PE teachers used the Dutch handbook of PE [20] although teachers have great autonomy to decide what to do in their classes. Further research to improve our insights into children’s GMC developmental trajectories in one school year should include other factors such as sport participation, social-economic status, outdoor play opportunities, and quality of PE classes.

## 5. Conclusions

To conclude, this study elucidates that, on average, children develop their GMC in one school year. However, we found differences between children’s individual developmental GMC trajectories and a substantial number of children did not show any positive GMC development. These differences in development are associated with children’s grade and baseline proficiency level. Those with lower proficiency levels do develop GMC during a school year but may risk a fallback during the summer break. Further research should examine GMC development with more possible influencing factors as well as over a longer time span to better understand differences in children’s GMC development. This may result in more individualized programs in PE lessons, enabling children to optimally develop their GMC, and a better use of GMC assessment tools to monitor children’s development.

## Figures and Tables

**Figure 1 children-09-00964-f001:**
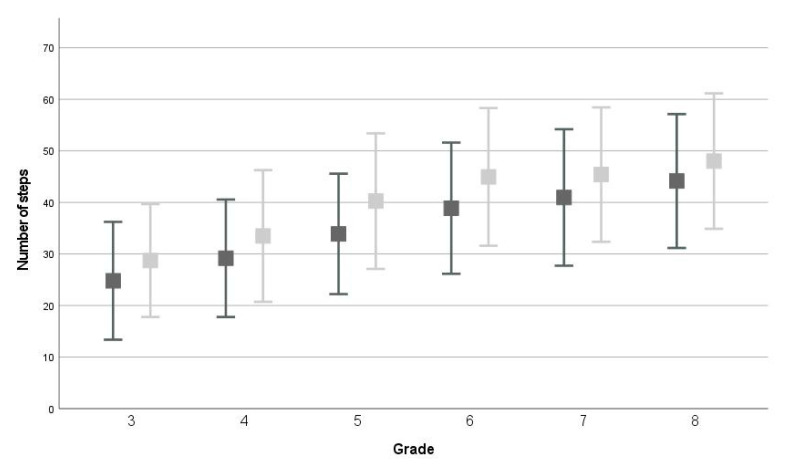
Average and standard deviation on walking backwards test at T0 (dark grey) and T1 (light grey).

**Figure 2 children-09-00964-f002:**
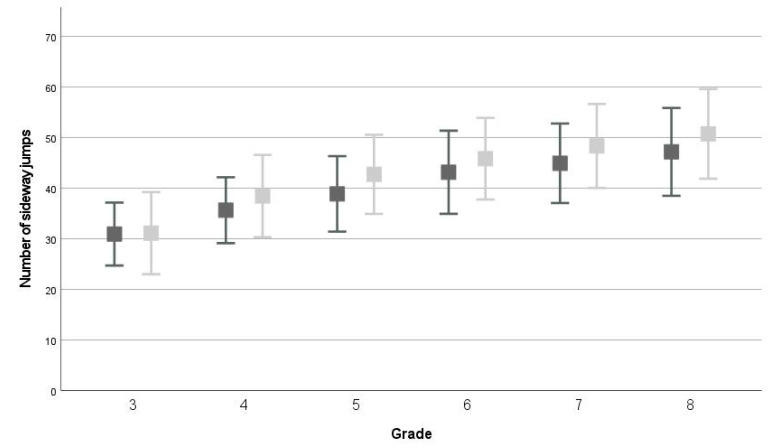
Average and standard deviation on jumping sideways test at T0 (dark grey) and T1 (light grey).

**Figure 3 children-09-00964-f003:**
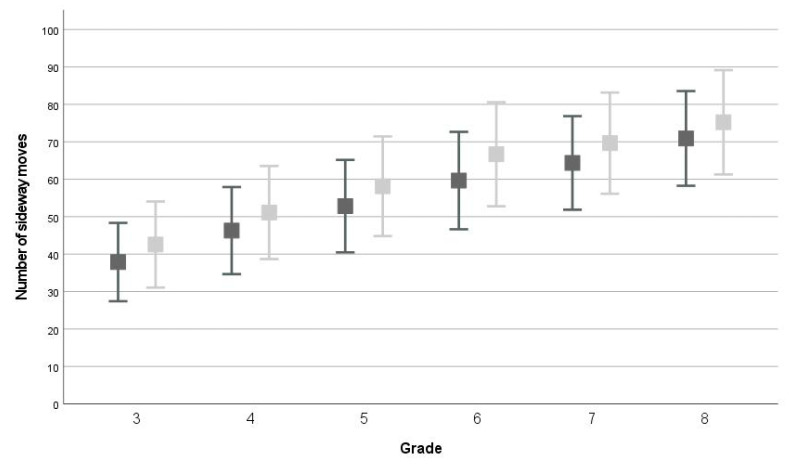
Average and standard deviation on moving sideways test at T0 (dark grey) and T1 (light grey).

**Figure 4 children-09-00964-f004:**
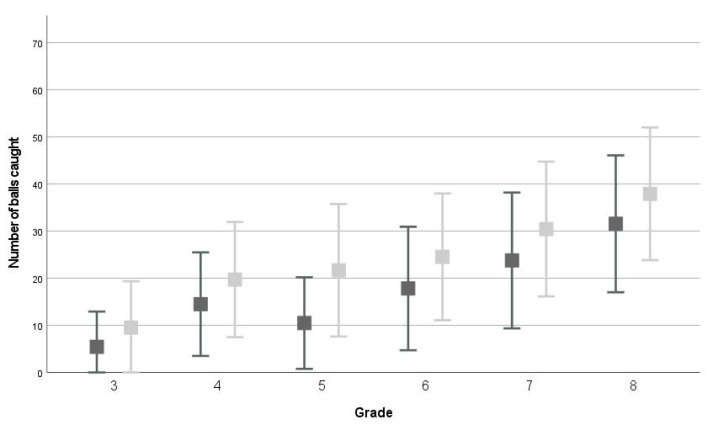
Average and standard deviation on eye hand coordination test at T0 (dark grey) and T1 (light grey).

**Figure 5 children-09-00964-f005:**
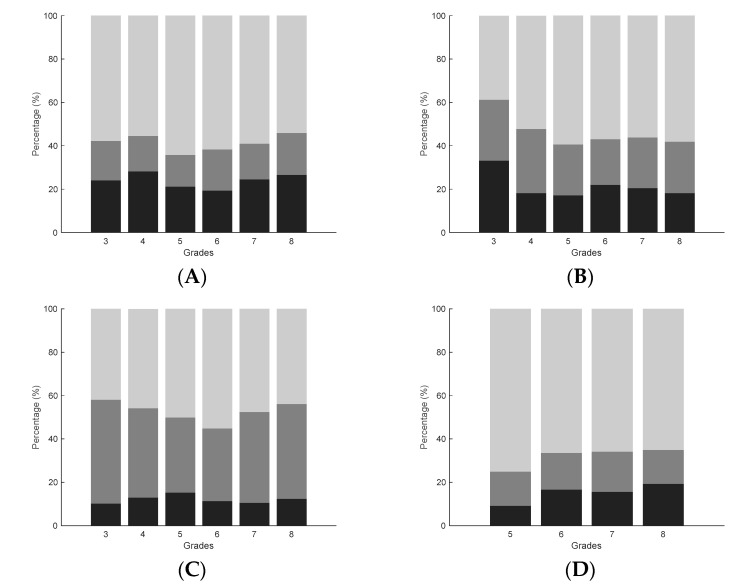
Percentages of children with positive (light grey), no (dark grey), or negative (black) development per grade. Results per panel: (**A**) walking backwards, (**B**) jumping sideways, (**C**) moving sideways, (**D**) eye hand coordination.

## Data Availability

Not applicable.

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
