# Peer review of "Primary School Pupils: Unequal GMC Developmental Pathways in a Single School Year"

_children, 2022, doi:10.3390/children9070964_

Round 1

Reviewer 1 Report

The manuscript presents a relevant topic to publish in Children Journal, which could be accepted with some minor revisions.

In my opinion, the introduction provides adequate information and structure to set up the research questions raised in the manuscript; the methodology provides sufficient detail, but that can still be an improvement; the results section is sufficiently clear and precise; the discussion and conclusion are coherent based on the results of the study and the previous literature.

After carefully reading your manuscript, I point out some aspects that must be improved and corrected:

- The authors should improve the methodology section about the description of the sample and the motor assessment instrument;

- How was the sample constituted? Is it a random sample? What were the inclusion/exclusion criteria? These criteria are vague (“Participants had neither inju-109 ries nor other issues that could hinder their performance“- lines 109-110.). Please provide more information about the participants;

- Regarding the KTK instrument, the authors should describe some of its main psychometric properties. Is the instrument validated for Dutch children?

- who evaluated the children? How was the evaluation of 2594 children carried out? I think the authors will be able to describe this process better;

- the manuscript does not report information about the quotation of the tests. What is the minimum or maximum test score? What cut-off values define whether the child has a positive or negative development / or a low, medium or high motor proficiency? I think this information should be included in section 2.2 of the manuscript to understand better the data reported in the tables and figures;

- I think the label of figures 2, 3 and 4 of the y axis is incorrect;

- All statistical symbols must be in italics (n, p, …. );

- Some formatting aspects should be corrected (spelling, punctuation). Please, correct what is pointed out in the body of the manuscript;

Reviewer 2 Report

Thank you for allowing me to review this paper, which is well written and deals with important issues on motor development.

All my comments and suggestions can be found in the attached pdf

Reviewer 3 Report

This was a very interesting paper, you managed to gain a very good sample size. You highlighted how those children with lower FMS proficiency flourish in the PE environment, which is brilliant, however, the detriment to children entering school with a higher FMS proficiency is a concern. Well done on highlighting this please see my comments attached.
